# Dual Use of Public and Private Health Care Services in Brazil

**DOI:** 10.3390/ijerph19031829

**Published:** 2022-02-06

**Authors:** Bianca Silva, Niel Hens, Gustavo Gusso, Susan Lagaert, James Macinko, Sara Willems

**Affiliations:** 1Department of Public Health and Primary Care, Faculty of Medicine and Health Sciences, Ghent University, BE-9000 Ghent, Belgium; Susan.Lagaert@UGent.be (S.L.); sara.willems@ugent.be (S.W.); 2Data Science Institute (DSI), I-BioStat, Hasselt University, BE-3500 Hasselt, Belgium; niel.hens@uhasselt.be; 3Faculty of Medicine, University of São Paulo, São Paulo 05403-000, Brazil; gustavo.gusso@usp.br; 4Department of Health Policy and Management, University of California Los Angeles, Jonathan and Karin Fielding School of Public Health, Los Angeles, CA 90095, USA; jmacinko@g.ucla.edu

**Keywords:** healthcare use, Brazil, public health, private healthcare

## Abstract

(1) Background: Brazil has a universal public healthcare system, but individuals can still opt to buy private health insurance and/or pay out-of-pocket for healthcare. Past research suggests that Brazilians make combined use of public and private services, possibly causing double costs. This study aims to describe this dual use and assess its relationship with socioeconomic status (SES). (2) Methods: We calculated survey-weighted population estimates and descriptive statistics, and built a survey-weighted logistic regression model to explore the effect of SES on dual use of healthcare, including demographic characteristics and other variables related to healthcare need and use as additional explanatory variables using data from the 2019 Brazilian National Health Survey. (3) Results: An estimated 39,039,016 (*n* = 46,914; 18.6%) persons sought care in the two weeks before the survey, of which 5,576,216 were dual users (*n* = 6484; 14.7%). Dual use happened both in the direction of public to private (*n* = 4628; 67.3%), and of private to public (*n* = 1855; 32.7%). Higher income had a significant effect on dual use (*p* < 0.0001), suggesting a dose–response relationship, even after controlling for confounders. Significant effects were also found for region (*p* < 0.0001) and usual source of care (USC) (*p* < 0.0001). (4) Conclusion: A large number of Brazilians are seeking care from a source different than their regular system. Higher SES, region, and USC are associated factors, possibly leading to more health inequity. Due to its high prevalence and important implications, more research is warranted to illuminate the main causes of dual use.

## 1. Introduction

It is common to find a combination of public and private healthcare systems in both low- and middle-income countries (LMICs) and high-income countries [1,2,3]. In LMICs, such a combination often entails underfunded public care and restricted access to private providers. Such arrangements are commonly found in Latin American countries [4], Middle Eastern and North African countries [5,6], among others [7,8]. Some research points to private services in LMICs having lower quality than public services, even when considering the formal and informal sector separately [9], while others show better indicators in the private sector [10]. In the debate about universal health coverage (UHC), involving the private sector in many different arrangements has been proposed as a solution for government cost containment [11].

Public healthcare in Brazil exists in the form of a Beveridgian tax-financed, free at the point of care, universal system named the Sistema Único de Saúde—Unified Health System (SUS), created in 1988 with the writing of a new constitution at the end of the military dictatorship [12]. In 2013, SUS was responsible for 61.7% of medical visits [13] and, in 2019, 64.6% of hospital admissions in the country were paid by SUS [14]. SUS also has a strong primary care program, the Family Health Strategy (FHS), which covers 62.6% of the Brazilian population [15], along with a comprehensive national vaccination program and essential medicines list, which are both offered for free [16,17].

Private healthcare in Brazil consists of private out-of-pocket services and a large private health insurance market, with over 48 million users, or 24.9% of the Brazilian population, but a higher share of the total health expenditure in the country [16,18,19]. According to the Organization for Economic Co-operation and Development (OECD) classification of the role of private health insurance in a healthcare system, the Brazilian model of private plans and services would be classified as a duplicate and supplementary model. This means people can buy private plans with the possibility of having easier access to certain services or technologies, or to different facilities or professionals [20]. Private insurance plans in Brazil may have different formulas, including monthly premiums, copayments, etc., and different coverages ranging from ambulatory care only, without lab or hospital coverage, to very high coverage, including elective aesthetic surgeries, but most of them have limited geographical coverage [21,22,23]. Since healthcare is a constitutional right in Brazil and SUS is a universal system, private plan holders are not excluded from it.

Despite the theoretical ideological separation of these two systems, they are, in practice, interconnected in many ways. The state offers, for example, subsidies for private health insurance providers and tax waivers for private plan users and employers who offer them as job benefits, and government institutions often offer private health insurance for their civil servants [16,24,25]. Additionally, many SUS hospitals, diagnostic services, and even primary care units are offered or managed by private (or nonprofit) organizations contracted by the government, limiting the initial idea of a fully state-owned healthcare. Defenders of private health insurances argue that they relieve the pressure on SUS by absorbing part of the demand and allowing it to focus its scarce resources [26]. However, private health insurances in Brazil rarely cover medication or vaccination costs, many plans offer incomplete coverage for hospital admissions or more complex procedures, or simply do not clearly describe their coverage [23,27].

Previous research has shown persistent regional and socioeconomic inequities in healthcare in Brazil. For instance, the north and northeast regions of the country have lower healthy life expectancy [28], but also less access to healthcare [29]. Higher income or education has also been associated to higher healthcare utilization, with having a usual source of care (USC), with seeking preventive care, and with higher private health insurance ownership [30,31,32,33,34]. Additionally, private insurance owners seek care more often than those that do not own insurance [35]. However, public health strategies, such as the FHS and the More Doctors Program (Programa Mais Médicos, a government program created to attract doctors to isolated or underserved areas in the country), have appeared to reduce the gap in utilization of dental and medical visits between the rich and poor [34], increase access to care, and improve health outcomes [31,36,37].

Evidence of the interconnectedness of these two supposed separate systems from the perspective of healthcare use also exists. There are some open data on the use of public services by private plan holders, as well as legal mechanisms to make insurance companies pay private health plan holders’ use of SUS services back to the government [38]. Such data, however, are incomplete due to difficulties in identifying single users and focus mostly on emergency and hospital admissions, leaving out ambulatory care, among other procedures. A recent literature review indicated that private users may resort to SUS up to 13% of the times they seek care [39]. Nonetheless, estimating private spending by otherwise public users is difficult due to the lack of a database for out-of-pocket services. Patient itinerary studies seem to indicate that habitual users of the public system can seek private services through out-of-pocket payment for subspecialist care. On the other hand, private users can seek SUS for high-complexity treatments not covered by their plans, medications, and for homecare follow-up with public primary care [40,41,42].

This dual use of healthcare has the potential to accentuate health inequities, since those able to do it would be the ones of a higher socioeconomic status (SES) who already have better access to care. It may also lead to redundant use of resources when public and private healthcare payers are both financing and offering the same services to the same people. This study aims to quantify and describe dual use from the perspective of the user, who can navigate the system both from the direction of private insurance owners using public services, as well as from the direction of people who do not own private insurance seeking private services, and assess its relationship with SES, along with other healthcare use variables and confounders.

## 2. Materials and Methods

This study used data from the 2019 Brazilian National Health Survey (Pesquisa Nacional de Saúde—PNS) [43] to assess personal affiliation to different healthcare systems and identify dual users of healthcare. Descriptive analyses of the data were performed, taking into consideration the complex sample design [44] to calculate total population estimates and survey-weighted distributions. We built a survey-weighted logistic regression model to explore the association of SES with dual use of healthcare, including demographic characteristics and other variables related to healthcare need and use as additional explanatory variables. The PNS survey was administered in the period between August 2019 and March 2020 and is currently the country’s most recent National Health Survey.

### 2.1. Survey Design, Sampling, and Fieldwork

The PNS 2019’s target population consisted of a nationally representative sample of noninstitutionalized individuals selected from permanent housing [44]. A stratified sampling technique with three clustering stages was used. The primary sampling units were the census tracts, the secondary units were the households, and the third-stage units were residents aged 15 or older. A total of 108,525 household visits were planned, with an expected nonresponse rate of 20%. In total, 100,541 were visited and 94,114 interviews were completed, resulting in a nonresponse rate of 13.2% and an excess coverage of 7.3% [45]. In these households, one resident aged 18 or older responded to questions about the household (assets, electricity, water and sanitation, etc.) and about demographic and health information for each resident. Additionally, one resident older than 15 years in each household was randomly selected for an individual interview. The complete PNS questionnaire can be found at the website of the Brazilian Institute of Geography and Statistics (IBGE). The interviews were conducted face to face with trained interviewers equipped with a handheld computer. The authors did not participate in the data collection process. Interviews were voluntary and no financial incentives were provided. The PNS project was approved by the Brazilian National Research Ethics Committee/National Health Council and approved as per Opinion No. 3.529.376, issued on 23 August 2019. The de-identified data are available to the public on the IBGE website [43,44].

### 2.2. Definition of Dual Use

To define dual use of healthcare, the questions on health insurance ownership and the source of care for the last care episode were used. A binary variable was built, which was true when the individual had no private health insurance but a private source was indicated for the last care episode, or when the individual had private health insurance but a public source was indicated for the last care episode. The dual use variable was false when the individual had no private health insurance and used a public source for the last care episode, or the individual had a private health insurance and used a private source for the last care episode. For the last care episode, responses were restricted to those with a clear link to the private or public system. The options where this was unclear, such as “pharmacy” or “home care”, were excluded.

### 2.3. Variable Coding

Variables of interest were selected based on the literature and used as explanatory variables [46]. These included the following demographic variables and those related to healthcare use and need: sex coded as “male” and “female”; age groups coded as 0–14, 15–24, 25–34, 35–44, 45–54, 65–74, and 75+ years of age; race/skin color coded as white and non-white; living in a rural or urban dwelling; geographic region coded as “North”, “Northeast”, “Southeast”, “South”, and “Center-west”; self-reported health status coded as good/very good, fair, and bad/very bad; and household enrollment in the FHS coded as “yes” and “no/does not know”. USC was assessed by combining the answers to the questions “does ____ usually seek the same place, doctor or health care service when in need of care?” and “when sick or in need of health care, ____ seeks:” coded as “public primary care”, “public secondary care”, “public emergency care”, “public hospital outpatient care”, “private outpatient care”, “private emergency care”, “other”, and “no USC”.

SES was evaluated through three different variables: educational attainment, divided into “no education/primary incomplete”, “secondary incomplete”, “undergraduate incomplete”, and “undergraduate complete and above”; monthly income; and material wealth in quintiles. The latter was calculated based on an index generated by 17 variables of household assets and housing conditions. We used weighted principal component analysis to reduce the number of variables to one and generated a wealth index by extracting the first component of the model and splitting the resulting variable into quintiles [47]. Table 1 presents definitions for each variable.

### 2.4. Data Analysis

Since only people who sought care in the 2 weeks prior to the survey responded to the question on source of care for the last care episode, the population studied was a subset of the total population in the survey. In order to assess how different this subset was from the general population, a comparison of proportions was performed using a weighted chi-square test with the modification proposed by Rao and Scott for survey analysis including the demographic-, socioeconomic-, and healthcare-use-related explanatory variables previously mentioned. Similarly, a chi-square test was computed for bivariate analyses comparing the population of dual users vs. non-dual users.

Survey-weighted quasi-binomial logistic regression models were built to explain the outcome variable (dual use), and a type 3 analysis of variance (ANOVA) was performed to determine which variables had a significant effect on dual use. The variance inflation factor (VIF, which signals potentially problematic inflation of the standard errors due to multicollinearity problems) was calculated, showing low collinearity among the different covariates. The first model built had only the main effects, while the second model had three interaction terms to assess the effect of SES on women, non-white persons, and those enrolled in the FHS, effects deemed theoretically relevant. Finally, in order to discover new possible interactions, a third model was built via a combination of forward and backward model selection.

The following steps were used for variable selection for the third model: (1) socioeconomic and demographic variables, as well as variables indicated in the literature to have an effect on healthcare use, were included; (2) separate models were built including all main effects, plus an interaction term of each pair of two variables; (3) the variable pairs that achieved a level of significance below *p* < 0.05 were selected; (4) a model including main effects and the significant interaction terms was built; (5) interaction terms that were no longer significant were dropped from the model until all interaction terms included were significant. To ensure accounting for possible confounders, interaction terms indicated as plausible by the literature were retained regardless of achieving significance. Given the large sample size being studied, as well as the many statistical tests performed, we opted for stricter significance thresholds in order to avoid type I errors [48]. A Bonferroni correction to the α level of significance was performed, and the family-wise error rate (FWER) was calculated at 0.09%. P values are reported, 99% confidence intervals (CI) were calculated for the model coefficients, and standard errors (SE) were calculated for all survey-weighted percentages. Only survey-weighted percentages are reported.

The analysis was made using the following software: R Core Team, R software version 4.0.3 (R Foundation for Statistical Computing, Vienna, Austria); RStudio version 1.3.1093 (PBC, Boston, MA, USA); and Lumley, T. “survey: analysis of complex survey samples”, R package version 4.0.

## 3. Results

Using the PNS 2019 complex survey design, we estimated that 39,039,016 (*n* = 46,914; 18.6%; SE = ±0.2) persons sought medical care in the two weeks prior to the survey, forming the subset analyzed in this study. The characteristics of this subset can be found in Table 2, which includes a comparison to the general population in the survey in relation to the variables being studied. The chi-square test computed to test for differences in the distribution of sample characteristics among the subset and general population was found to be significant for all variables studied, except for enrollment in an FHS unit.

In the two weeks prior to the survey, the population who sought care was mainly female (*n* = 29,290; 62%; SE = ±0.3) and had an over-representation of white individuals (*n* = 18,497; 47%; SE = ±0.6) and those of higher socioeconomic status (see Table 2). Despite the southeastern region of the country being more populated than other regions, the region was also over-represented in the subset (*n* = 11,324; 47.4%; SE = ±0.5), as was the southern region (*n* = 6145; 15.1%; SE = ±0.3). Finally, private usual sources of care and public emergency care were also over-represented.

### 3.1. Description of Dual Users vs. Non-Dual Users

Among those who sought care in the last two weeks, an estimated 5,576,216 (*n* = 6484; 14.7%; SE = ±0.3) fell into the definition of dual use. Table 3 shows the characteristics of these individuals according to the selected variables, including SES and healthcare use variables. It includes bivariate analyses with a chi-square test for associations. SES was represented by the variables “material wealth”, “income”, and “education”, which were found significantly associated with dual use in these first analyses (*p* values, respectively, 0.0238; >0.0001; and 0.0005).

Dual users were predominantly female (62.8%; SE = ±0.9), non-white (52%; SE = ±1.2), did not have health insurance (67.3%; SE = ±1.2), and were from the southeastern region of Brazil (45.2%; SE = ±1.2). Differences between race and gender were not statistically significant in bivariate analyses, whereas regional differences were (see Table 3). The majority sought care in private facilities, with 67.3% (SE = ±1.2) of visits versus 32.7% (SE = ±1.2) in public facilities. In absolute numbers, this means an estimated 3,762,141 (*n* = 4629) patients without private insurance sought care in private institutions, while another 1,824,075 (*n* = 1855) private insurance holders sought care in the public system. For the public system, the main type of facility sought for care were primary care facilities (*n* = 1091; 18.9%; SE = ±0.9), while, for the private system, outpatient care was identified as the main source of care (*n* = 4510; 65.3%; SE = ±1.2) (see Table 3).

The act of migrating from one healthcare payer to another seems to increase with both material wealth and income quintiles, reaching its highest points at the fourth quintiles for both variables (see Table 3 and Figure 1). The same pattern is not observed for educational levels, with most dual users belonging to the category “no education/primary in-complete” (*n* = 2554; 40%; SE = ±1.0), followed by the category “undergraduate incomplete” (*n* = 1877; 31%; SE = ±1.0). The age distribution among dual users vs. non-dual users showed an over-representation of individuals 45 and older, which was statistically significant (dual users 45–54: 17%, SE = ±0.8; 55–64: 15.8%, SE = ±0.8; 65–74: 11%, SE = ±0.6; +75: 8.5%, SE = ±0.6; non-dual users 45–54: 15.7%, SE = ±0.3; 55–64: 14.7%, SE = ±0.3; 65–74: 10;4%, SE = ±0.3; +75: 6.6%, SE = ±0.2; *p* < 0.0001) (for full overview, see Table 3).

Most dual users referred to no usual source of care (*n* = 1870; 27.9%; SE = ±1.0), followed by public primary care as their USC (*n* = 1744; 27%; SE = ±1.0), and private outpatient care (*n* = 1499; 23.2%; SE = ±1.0). In turn, most non-dual users referred to public primary care as their usual source of care (*n* = 15,070; 36.1%; SE = ±0.7), followed by private outpatient care (*n* = 7842; 22.3%; SE = ±0.6), and no usual source of care (*n* = 7629; 18.5%; SE = ±0.4) (for full overview, see Table 3). These differences were statistically significant. Other variables investigated, such as urban or rural dwelling, self-assessed health status, and enrollment with a Family Health Strategy unit, were not found to have significant differences between dual users vs. non-dual users (see Table 3).

### 3.2. Multinomial Regression Analysis

We built three logistic regression models to explore the effects of SES, healthcare use, and demographic variables with dual use and performed a type 3 ANOVA to determine which variables had a significant effect on dual use. Table 4 presents the ANOVA table for the first model, with main effects only, in which usual source of care, income, and region achieved the 0.09% level of significance according to the Bonferroni-corrected FWER. Table 5 shows the in-group comparisons for the aforementioned variables, along with their respective odds ratio and Bonferroni-corrected *p*-values. Only significant differences were included in the table. Table 6 presents for the same model the odds ratio (OR) and a 99% confidence interval (CI) for each level of the variables in relation to the reference category. The results for the two other models built including interaction terms can be found in Appendix A.

In the analysis, the higher the income, the higher the OR of being a dual user, up to the fourth quintile (OR: 2.43; 99% CI: 1.91–3.08), with a slight decline on the fifth quintile (OR: 1.69; 99% CI: 1.26–2.26). The third and fourth quintiles had higher odds of dual use when compared to the second quintile, whereas the fifth quintile had lower odds than the third and fourth quintile (see Table 5). A similar pattern when comparing each level to the lowest quintile was found for material wealth. Living in the southeast (OR: 0.79; 99% CI: 0.64–0.99) or center-west (OR: 0.78; 99% CI: 0.63–0.98) regions had a negative effect, reducing dual use of the healthcare system in relation to living in the north region, and also when compared to all other regions (see Table 5; Table 6).

Several comparisons between usual sources of care were found to be significant. Having a public primary care unit as one’s usual source of care had a negative effect on dual use when compared to “other” (OR: 0.683; 99% CI: 0.45–1.04), as did having a private emergency care facility as the usual source of care (OR: 0.406; 99% CI: 0.23–0.72). Indicating public emergency care or a public outpatient hospital as one’s USC increased the odds of dual use when compared to public primary care (OR: 1.42 and 1.64, respectively). Private emergency care as a USC reduced dual use when compared to public primary care, public secondary care, public emergency care, public outpatient hospital, and private outpatient care. Finally, having no USC increased the odds of dual use when compared to all other levels, except for “other”, “public secondary care”, and “public outpatient hospital care”. For a full overview of comparisons, see Table 5.

## 4. Discussion

The purpose of this study was to describe dual use of healthcare in Brazil and to investigate its relationship with SES. Dual use was found to happen both on the direction of public to private and from private to public, although the former is more common than the latter. The findings confirm the hypothesis that higher SES, in particular, income, is associated with dual use of care, even after controlling for other variables and correcting for over-testing. Additionally, the regression analysis showed a significant effect of region and of usual source of care on the outcome.

This study looked at the dual use of public and private healthcare services from the perspective of the user. In our definition, a dual user is someone that is affiliated to a certain healthcare system (public or private) seeking care in a service that is part of the other system. In Brazil, as in many other parts of the world, there are private healthcare facilities offering care via the public system, especially for imaging and diagnostic tests, being financed by the government through contracts [29]. This was not the object of this study. In the Brazilian case and from the perspective of users, such facilities are still perceived as public care, since no direct payments need to be made at the point of care. Additionally, in the questionnaire used in the PNS survey, the question referring to the last care episode had as possible answers facilities that clearly belonged to either one or the other system [43]. The options where this was less clear were excluded from the analysis.

Our sample showed that women seek care more often than men. This is consistent with other studies in Brazil [30,32,35]. Reasons for women seeking care more often than men can be related to stronger preventive messages targeting women, which can be linked to the medicalization of the female body [49]. Additionally, patriarchal gender identities may lead to social expectations for women to seek (preventive) care and for men to postpone it [50]. Such differences were neither significantly intensified nor reduced among dual users.

In our sample, persons aged 75 years or older had a 1.49 times higher chance of dual use of healthcare than the reference group. Past research on the demographic traits associated with healthcare use has shown age as an important factor, with the lowest and the highest age groups being associated with more frequent use of healthcare [32]. This is probably due to the higher health needs in both extremes of life.

Living in the southeast or the center-west region had a significant negative effect on dual use of healthcare when compared to all other regions except each other. Rural dwellings had higher odds of dual use. Many past studies have shown the persistent regional inequalities in Brazil, as well as inequalities between rural and urban settlements [29,32,51]. These span from economic and health inequalities between regions [28] to unequal development of the healthcare system and distribution of medical facilities among the regions, which privilege the southeast region [13]. Previous research has found that lower use of SUS in certain regions was associated with higher private health insurance ownership [35,39]. The southeastern region has the lowest FHS coverage in the country [15] and the highest private health insurance ownership [33]. FHS coverage in the center-west region varies greatly between states [15], and its private health insurance ownership rate is slightly higher than the national rate [33]. Possible explanations for our results would be (a) private health insurance with better service coverage makes private insurance owners more loyal to private care; and (b) well-developed primary care in some center-west states and/or secondary care in the southeast has led to more loyalty to public care. Dual use seems to be related to the availability and perceived quality of healthcare services either in the public or in the private system, the purchasing power of the user, and possibly to the user’s own perception of their health needs, which is also affected by sociocultural factors. Further research is necessary to clarify these underlying mechanisms in the different regions of the country.

In our sample, FHS enrollment was not found to be related to seeking care in the last two weeks. Similarly, it did not have a significant effect on dual use. FHS, the main Brazilian approach to primary care, consists of a multi-professional healthcare team, including at least a general practitioner, a nurse, and a group of community health workers [52]. These teams are responsible for a patient clientele that is geographically determined, and the community health workers must both come from this community and visit patients at least monthly [52]. The FHS has been shown to improve access to care and to be associated with having a usual source of care [30,31]. Additionally, a previous study found that FHS enrollment was associated with increasing the use of SUS and lowering the use of private services by those without a private health insurance, and with increasing the use of SUS by private plan holders [53]. Differences between our results and previous research may be explained by the different study designs. Our analysis looked specifically at dual use, whereas previous analyses explored the effect of these variables on any use of healthcare. Additionally, FHS coverage is notably higher in the lower socioeconomic strata [15], which may naturally not have the economic means to access private healthcare services.

In our sample, health status had a strong association with seeking care in the last two weeks, but persons reporting their health to be “bad” or “very bad” were less likely to use dual care when compared to those reporting “good” or “very good” health. Having worse health status has been typically associated with higher use of healthcare, due to higher health needs [34]. This might lead to a stronger relationship with their healthcare provider, increasing loyalty. The implication is, however, that those with self-reported good health have higher odds of seeking dual use of care when compared to those with self-reported bad health, raising the possibility of over-utilization.

Individuals identifying a public primary care unit or a private emergency service as a USC had a lower chance of dual use when compared to several other options, while having no USC increased dual use. Having a usual source of care is a measure of the availability of healthcare, but also a possible indicator of continuity of care. Primary care units have been indicated to be the main USC for Brazilians, as well as the main source of last care in previous research [14,31,54]. Healthcare systems where primary care is strong often have better health outcomes [55,56]. Our results may suggest good quality of care in the Brazilian primary care units, preventing dual use. However, it is also possible that persons of less financial means report primary care units as their usual source of care while also not having the financial means to seek private care if they deem it necessary. The contradictory nature of the two explanations warrants further research on this topic.

Our research findings highlight the association of SES with both healthcare use and dual use of care. Income had the strongest association, followed by material wealth, whereas education did not achieve significance. This may mean that dual use is mainly determined by the purchasing power one may have. In the highest quintile, there is a decrease in dual use, which could be related to having access to private health insurances with better coverages [33].

The dose–response relationship tendency found between income and dual use raises questions about health inequity and over-utilization of health care. Previous research has shown that the lower the SES, the higher the healthcare need. Nonetheless, the “Inverse Care Law” [57] postulates that most health care services will be concentrated where there is the least need for healthcare. This has implications for, on the one side, the unmet need of those with lesser means, but, on the other side, the possibility of unnecessary interventions, treatments, and expenditures for the well-off [58].

Another important finding was that the main facilities sought by dual users were private outpatient care and public primary care. This may be evidence of the weak points in both systems in Brazil. Specialist care is a well-known bottleneck in SUS, being mentioned in some studies as a reason for seeking private care [40,42]. Nonetheless, private healthcare in Brazil has a near absence of organized primary care services. Private insurance owners seek care directly with specialists, often choosing a different specialty for every different health-related problem they may have, with some seeking primary care in SUS [40]. Previous research has shown that private users sometimes seek SUS for high-complexity care as well [42]. Although the percentage of public hospital outpatient care dual use in our results is low when compared to other sources of care, such care is more expensive and often the driver of high costs in healthcare systems [59].

One possible implication of dual use not addressed in this study is the fragmentation of care. Referral systems in Brazil are weak and communication between different levels of care in the public system can be very impersonal and bureaucratic or simply nonexistent [60,61,62]. There are no studies on referral systems in private care in Brazil. Communication between public and private providers is likely to be even more difficult than among providers of the same system, with important implications for co-ordination of care and health outcomes.

The interaction terms included in the second model (see Appendix A, Table A1 and Table A2) showed significant effects of income on women, persons of non-white race, and those enrolled with the FHS. The forward and backward model selection procedure also identified new significant interactions, including USC and education, and USC and income, among other potential interactions (see Appendix A, Table A3 and Table A4). This suggests new possibilities for research in the area, exploring how the place where one usually receives care can influence their care-seeking behavior, mediated by one’s SES. However, the inclusion of many interaction terms hampers the interpretation of the main effects. In order to improve the interpretability and highlight the robustness of our results, we opted to include these models in the Appendix A, leaving them out of the main report.

This study has several potential limitations. Firstly, the subset used to investigate dual use, namely the population that sought care in the last two weeks, was significantly different than the general population. Although this may indicate a loss of generalizability, chi-square tests are extremely sensitive to sample sizes, which may lead to small differences being detected as significant. The problem of working with too large of a sample was addressed in the analysis of variance by also reporting the Bonferroni-corrected FWER [48]. Secondly, the definition of dual use was created based on convention in the Brazilian literature for defining “SUS-dependent population” as those who do not own private health insurance, notwithstanding the deserving criticisms that such a construction has received [63]. However, there may be a small group of wealthier persons who do not own health insurance but always pays out of pocket for private healthcare and would not consider themselves as users of SUS. Additionally, the fact that our data are limited to care-seeking behavior in the last two weeks reduces the reach of the results presented, which could have been substantially higher had the question considered source of care in the past 12 months. Still related to the data collection procedure, one interviewer was responsible for the entire interview, which meant the ones evaluating subjective components were not masked from other aspects of the status of the participants. Finally, the variables in this survey are based on information offered by one household informant, which may be subject to recall bias.

Despite these limitations, the present study is, to the best of our knowledge, the first to measure the magnitude of dual use. It has found that a large number of Brazilians are seeking care in a source different than their regular healthcare system. Higher SES, region, and USC are associated with dual use, even when controlling for possible confounders. This phenomenon may be linked to poor co-ordination and quality of care, overutilization, waste of resources, and aggravation of health inequities. Due to the high prevalence of dual use and its important implications, more research is warranted to help illuminate the main roots of this problem. Future research could address the main reasons healthcare users seek care in a different system and what is their experience when making dual use of healthcare. Both are likely to be different for the users migrating from public to private services compared to those migrating from private to public services.

## 5. Conclusions

The Brazilian public health system has been groundbreaking in expanding its coverage in such a large territory and investing in primary and community care. It is an example for many low- and middle-income countries in its offer of universal care, comprehensive vaccination, and essential medicines programs. Such a successful system cannot afford to lose the ambition of fighting health inequities and of offering qualitative care in all different levels of the system and regions of the country. This research has important insights for policymakers about how the population uses healthcare in practice. The results presented here challenge directly the traditional ideas of clearly divided public and private healthcare in Brazil. The implication that both systems are interdependent but also mutually flawed should lead to innovative ways of thinking of solutions for universal health coverage in Brazil. One that understands duplicate coverage may be an unwanted consequence, both for patients as for healthcare payers.

## Figures and Tables

**Figure 1 ijerph-19-01829-f001:**
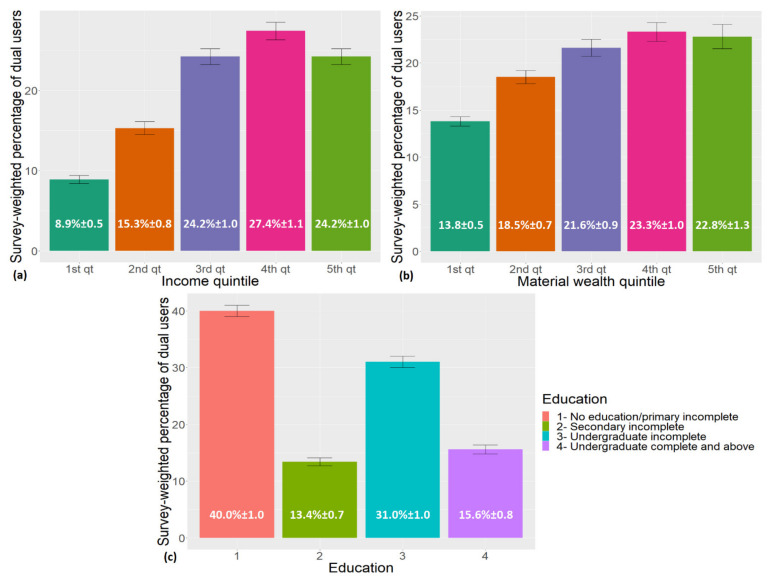
Survey weighted distribution of dual use across (**a**) income, (**b**) material wealth, and (**c**) educational attainment including standard errors.

**Table 1 ijerph-19-01829-t001:** Variable definitions.

Variable	Definition
Dual use	True: individual without private health insurance who used a private source for the last care episode individual with private health insurance who used a public source for the last care episode
	False: individual without private health insurance who used a public source for the last care episode individual with private health insurance who used a private source for the last care episode
Sex	Male
	Female
Age groups (in years of age)	0–14
	15–24
	25–34
	35–44
	45–54
	65–74
	75+
Race/skin color	White
	Non-white
Dwelling	Rural
	Urban
Geographic Region	North
	Northeast
	Southeast
	South
	Center-west
Self-reported Health Status	Good/very good
	Fair
	Bad/very bad
Household Enrollment in the FHS	Yes
	No/does not know
USC	Public primary care
	Public secondary care
	Public emergency care
	Public hospital outpatient care
	Private outpatient care
	Private emergency care
	Other
	No USC
Educational attainment	No education/primary incomplete
	Secondary incomplete
	Undergraduate incomplete
	Undergraduate complete and above
Monthly income	Quintiles
Material wealth	Quintiles

**Table 2 ijerph-19-01829-t002:** Description of population who sought care in the 2 weeks prior to the survey and general population in the PNS 2019.

Variable	Respondents Who Sought Care in the 2 Weeks Prior to the Survey			All			*p* Value
	N	WN	% ± SE	N	WN	% ± SE	
Total	46,914	39,039,016	100 (18.6 ± 0.2)	279,382	209,589,607	100	--
Dwelling							<0.0001
Urban	38,027	34,514,587	88.4 ± 0.3	174,259	144,517,015	84.7 ± 0.2	
Rural	8887	4,524,429	11.6 ± 0.3	58,209	26,033,576	15.3 ± 0.2	
Sex							<0.0001
Male	17,624	14,816,032	37.9 ± 0.3	116,818	85,342,077	50.0 ± 0.1	
Female	29,290	24,222,984	62.0 ± 0.3	115,650	85,208,514	50.0 ± 0.1	
Race							<0.0001
White	18,497	18,350,812	47.0 ± 0.5	80,522	72,686,910	42.6 ± 0.3	
Non-white	28,410	20,683,376	53.0 ± 0.5	151,929	97,854,237	57.4 ± 0.3	
Age							<0.0001
0–14	6920	5,653,355	14.7 ± 0.3	48,018	33,091,415	19.6 ± 0.2	
15–24	5036	4,037,134	10.5 ± 0.2	38,791	27,289,832	16.2 ± 0.1	
25–34	5513	4,435,939	11.6 ± 0.3	34,341	25,569,937	15.1 ± 0.1	
35–44	6738	5,693,580	14.8 ± 0.3	34,722	25,852,008	15.3 ± 0.1	
45–54	7147	6,086,616	15.9 ± 0.3	29,070	22,338,009	13.2 ± 0.1	
55–64	6705	5,700,443	14.9 ± 0.3	22,845	17,627,126	10.4 ± 0.1	
65–74	4819	4,007,847	10.4 ± 0.2	13,814	10,495,985	6.2 ± 0.1	
+75	3116	2,714,413	7.1 ± 0.2	8437	6,576,903	3.9 ± 0.1	
Material wealth							<0.0001
1st quintile	21,894	6,258,532	16.0 ± 0.2	124,048	35,659,389	21.0 ± 0.2	
2nd quintile	10,318	7,161,698	18.3 ± 0.3	49,094	34,754,593	20.4 ± 0.3	
3rd quintile	6789	8,110,930	20.8 ± 0.4	28,734	33,806,841	19.8 ± 0.3	
4th quintile	4695	8,276,514	21.2 ± 0.5	18,840	33,646,716	19.7 ± 0.3	
5th quintile	3218	9,231,342	23.6 ± 0.7	11,752	32,683,053	19.2 ± 0.4	
Income							<0.0001
1st quintile	9236	6,062,530	15.7 ± 0.4	60,624	34,924,549	20.7 ± 0.3	
2nd quintile	8956	6,903,809	17.8 ± 0.4	49,755	34,076,163	20.1 ± 0.3	
3rd quintile	9203	7,882,833	20.4 ± 0.4	43,964	34,169,054	20.2 ± 0.3	
4th quintile	8595	8,101,195	20.9 ± 0.4	38,708	33,722,160	20.0 ± 0.3	
5th quintile	10,500	9,752,815	25.2 ± 0.6	37,087	32,045,680	19.0 ± 0.3	
Education							<0.0001
No education/primary incomplete	19,355	15,598,252	43.0 ± 0.5	101,988	70,237,162	43.8 ± 0.3	
Secondary incomplete	5727	4,866,029	13.4 ± 0.3	32,837	24,683,737	15.4 ± 0.1	
Undergraduate incomplete	11,679	9,927,810	27.4 ± 0.4	59,363	46,330,722	28.9 ± 0.2	
Undergraduate complete and above	6749	5,857,799	16.2 ± 0.5	23,897	19,241,167	12.0 ± 0.2	
Region							<0.0001
North	8372	2,476,537	6.3 ± 0.1	53,075	15,648,213	9.9 ± 0.0	
Northeast	15,925	9,427,647	24.1 ± 0.3	83,628	47,522,444	27.9 ± 0.1	
Southeast	11,324	18,509,818	47.4 ± 0.5	45,016	69,877,525	41.0 ± 0.1	
South	6145	5,921,489	15.1 ± 0.3	25,093	24,021,083	14.1 ± 0.1	
Center-west	5148	2,703,525	6.9 ± 0.2	25,656	13,481,326	7.9 ± 0.0	
FHS							0.8701
Enrolled	30,866	24,408,374	62.5 ± 0.8	152,206	106,793,241	62.6 ± 0.5	
Not enrolled/does not know	16,048	14,630,642	37.5 ± 0.8	80,262	63,757,350	37.4 ± 0.5	
Health insurance							<0.0001
Yes	14,049	13,512,509	34.6 ± 0.6	44,548	41,075,465	24.1 ± 0.3	
No	32,865	25,526,507	65.4 ± 0.6	187,920	129,475,126	75.9 ± 0.3	
Health status							<0.0001
Very good or good	25,635	22,293,108	57.1 ± 0.4	173,046	130,710,571	76.6 ± 0.2	
Fair	15,856	12,529,640	32.1 ± 0.4	50,949	34,291,030	20.1 ± 0.2	
Bad or very bad	5423	4,216,268	10.8 ± 0.2	8473	5,548,990	3.3 ± 0.1	
Usual source of care							<0.0001
Public primary care	17,273	13,578,307	34.8 ± 0.6	90,192	61,561,878	36.1 ± 0.4	
Public secondary care	1104	926,077	2.4 ± 0.2	4236	3,260,442	1.9 ± 0.1	
Public emergency care	4286	3,869,702	9.9 ± 0.4	23,195	18,805,982	11.0 ± 0.3	
Public outpatient hospital care	2388	1,698,374	4.4 ± 0.2	12,833	8,420,497	4.9 ± 0.2	
Private outpatient care	9453	8,651,236	22.2 ± 0.5	32,254	28,000,797	16.4 ± 0;3	
Private emergency care	1720	1,805,915	4.6 ± 0.3	4975	5,254,904	3.1 ± 0.2	
No USC	9762	7,758,115	19.9 ± 0.4	59,523	41,435,080	24.3 ± 0.3	
Other	928	751,291	1.9 ± 0.1	5260	3,811,011	2.2 ± 0.1	

WN: weighted *n* in millions; %: within-group survey-weighted percentages; SE: standard error; *P* value of the survey-weighted chi-square test for comparison between the subset and general population.

**Table 3 ijerph-19-01829-t003:** Description and bivariate analysis of dual users vs. non-dual users in the PNS 2019.

Variable	Dual Users			Non-Dual Users			*p* Value
	N	WN	% ± SE	N	WN	% ± SE	
Total	6484	5,586,216	14.7 ± 0.3	39,161	32,464,691	85.3 ± 0.3	---
Source of care in the last care episode							---
Public	1855	1,824,075	32.7 ± 1.2	27,198	20,986,190	64.6 ± 0.7	
Public primary care	1091	1,056,345	18.9 ± 0.9	17,326	13,131,253	40.4 ± 0.6	
Public secondary care	165	164,875	3.0 ± 0.5	1975	1,581,826	4.9 ± 0.2	
Public emergency care	299	332,130	5.9 ± 0.6	4308	3,682,936	11.3 ± 0.4	
Public outpatient hospital care	300	270,725	4.8 ± 0.4	3589	2,590,176	8.0 ± 0.3	
Private	4629	3,762,141	67.3 ± 1.2	11,963	11,478,500	35.4 ± 0.7	
Private outpatient care	4510	3,649,346	65.3 ± 1.2	10,909	10,368,736	31.9 ± 0.7	
Private emergency care	119	112,795	2.0 ± 0.3	1054	1,109,765	3.4 ± 0.2	
Dwelling							0.8326
Urban	5302	4,936,837	88.4 ± 0.6	31,772	28,735,906	88.5 ± 0.3	
Rural	1182	649,379	11.6 ± 0.6	7389	3,728,784	11.5 ± 0.3	
Sex							0.4349
Male	2421	2,080,070	37.2 ± 0.9	14,684	12,330,217	38.0 ± 0.4	
Female	4063	3,506,146	62.8 ± 0.9	24,477	20,134,474	62.0 ± 0.4	
Race							0.3707
White	2,723	2,678,337	47.9 ± 1.2	15,346	15,205,573	46.8 ± 0.6	
Non-white	3,760	2.906,404	52.0 ± 1.2	23,809	17,255,764	53.1 ± 0.6	
Age							<0.0001
0–14	704	603,867	11.0 ± 0.6	6031	4,917,455	15.4 ± 0.4	
15–24	689	562,366	10.2 ± 0.6	4237	3,386,684	10.6 ± 0.3	
25–34	807	661,781	12.0 ± 0.6	4567	3,685,894	11.6 ± 0.3	
35–44	931	791,971	14.4 ± 0.7	5622	4,749,581	14.9 ± 0.3	
45–54	1035	938,279	17.0 ± 0.8	5915	4,995,966	15.7 ± 0.3	
55–64	980	872,479	15.8 ± 0.8	5563	4,695,428	14.7 ± 0.3	
65–74	730	604,638	11.0 ± 0.6	3967	3,303,700	10.4 ± 0.3	
+75	524	468,207	8.5 ± 0.6	2441	2,114,412	6.6 ± 0.2	
Material wealth							0.0238
1st quintile	2730	770,606	13.8 ± 0.5	18,473	5,290,345	16.3 ± 0.3	
2nd quintile	1546	1,031,484	18.5 ± 0.7	8536	5,960,294	18.4 ± 0.3	
3rd quintile	1039	1,206,076	21.6 ± 0.9	5590	6,708,142	20.7 ± 0.4	
4th quintile	733	1,302,748	23.3 ± 1.0	3855	6,787,801	20.9 ± 0.5	
5th quintile	436	1,275,302	22.8 ± 1.3	2707	7,718,109	23.8 ± 0.7	
Income							<0.0001
1st quintile	674	493,170	8.9 ± 0.5	8274	5,435,607	16.9 ± 0.4	
2nd quintile	1077	847,354	15.3 ± 0.8	7634	5,914,432	18.4 ± 0.5	
3rd quintile	1499	1,340,608	24.2 ± 1.0	7444	6,336,868	19.7 ± 0.4	
4th quintile	1638	1,515,270	27.4 ± 1.1	6735	6,359,361	19.8 ± 0.5	
5th quintile	1548	1,342,525	24.2 ± 1.0	8709	8,135,141	25.3 ± 0.6	
Education							0.0005
No education/primary incomplete	2554	2,125,761	40.0 ± 1.0	16,167	13,016,696	43.4 ± 0.5	
Secondary incomplete	799	710,452	13.4 ± 0.7	4793	4,057,513	13.5 ± 0.3	
Undergraduate incomplete	1877	1,649,275	31.0 ± 1.0	9519	8,040,664	26.8 ± 0.4	
Undergraduate complete and above	966	831,863	15.6 ± 0.8	5639	4,896,515	16.3 ± 0.5	
Region							0.0001
North	1001	329,207	5.9 ± 0.3	7036	2,050,520	6.3 ± 0.1	
Northeast	2130	1,360,014	24.3 ± 0.9	13,336	7,816,474	24.1 ± 0.4	
Southeast	1613	2,522,940	45.2 ± 1.2	9492	15,553,599	47.9 ± 0.5	
South	1007	1,022,744	18.3 ± 0.8	5011	4,759,052	14.7 ± 0.3	
Center-west	733	351,311	6.3 ± 0.3	4286	2,285,045	7.0 ± 0.2	
FHS							0.4941
Enrolled	4288	3,525,913	63.1 ± 1.2	25,687	20,215,961	62.3 ± 0.8	
Not enrolled/does not know	2196	2,060,303	36.9 ± 1.2	13,474	12,248,730	37.7 ± 0.8	
Health insurance							---
Yes	1855	1,824,075	32.6 ± 1.2	11,963	11,478,500	35.4 ± 0.7	
No	4629	3,762,141	67.3 ± 1.2	27,198	20,986,190	64.6 ± 0.7	
Health status							0.0691
Very good or good	3658	3,273,005	58.6 ± 1.0	21,355	18,514,420	57.0 ± 0.5	
Fair	2172	1,786,910	32.0 ± 0.9	13,249	10,411,676	32.1 ± 0.4	
Bad or very bad	654	526,302	9.4 ± 0.6	4557	3,538,595	10.9 ± 0.3	
Usual source of care							<0.0001
Public primary care	1744	1,504,077	26.9 ± 1.0	15,070	11,744,325	36.2 ± 0.7	
Public secondary care	148	151,909	2.7 ± 0.4	937	760,471	2.3 ± 0.2	
Public emergency care	595	562,753	10.1 ± 0;7	3577	3,207,923	9.9 ± 0.4	
Public outpatient hospital care	351	281,898	5.0 ± 0.4	1967	1,373,239	4.2 ± 0.2	
Private outpatient care	1499	1,294,930	23.2 ± 1.0	7842	7,254,370	22.3 ± 0.6	
Private emergency care	139	139,193	2.5 ± 0.3	1555	1,640,989	5.0 ± 0.3	
No USC	1870	1,559,743	27.9 ± 1.0	7629	6,005,851	18.5 ± 0.4	
Other	138	91,714	1.6 ± 0.2	584	477,522	1.5 ± 0.1	

WN: weighted *n* in millions; %: within-group survey-weighted percentages; SE: standard error; *P* value of the survey-weighted chi-square test for comparison between the subset and general population.

**Table 4 ijerph-19-01829-t004:** Analysis of variance table for the survey-weighted logistic regression model with the main effects of socioeconomic status on dual use of healthcare, controlling for demographic, health needs, and healthcare use variables—PNS 2019 (*n* = 41,921).

Variables	Degrees of Freedom	Chi-Square Test	*p* Value
(Intercept)	1	157.0421	<0.0001
FHS	1	2.7118	0.0980
Usual Source of Care	7	148.8832	<0.0001 ^a^
Sex	1	1.5545	0.2125
Age	7	16.9408	0.0177 ^c^
Race	1	0.4276	0.5132
Education	3	7.7329	0.0519
Income	4	125.2728	<0.0001 ^a^
Health Status	4	11.6643	0.0290 ^c^
Region	4	27.7091	<0.0001 ^a^
Dwelling	1	9.8556	0.0017 ^b^
Material Wealth	4	12.9843	0.0113 ^c^

^a^: *p* < 0.0009; ^b^: *p* < 0.01; ^c^: *p* < 0.05. Pseudo-R² (McFadden) = 0.13.

**Table 5 ijerph-19-01829-t005:** Post-hoc in-group comparisons of means for the variables “Region”, “Income”, and “Usual Source of Care”—Tukey contrasts.

Comparison	Odds Ratio	*p* Value *
Region		
Southeast—North	0.79	0.0590
Center-west—North	0.79	0.0548
Southeast—Northeast	0.78	0.0049
Center-west—Northeast	0.77	0.0040
South—Southeast	1.31	0.0016
Center-west—South	0.76	0.0054
Income		
2nd quintile—1st quintile	1.47	0.0002
3rd quintile—1st quintile	2.16	<0.0001
4th quintile—1st quintile	2.43	<0.0001
5th quintile—1st quintile	1.70	<0.0001
3rd quintile—2nd quintile	1.46	<0.0001
4th quintile—2nd quintile	1.65	<0.0001
5th quintile—3rd quintile	0.78	0.0442
5th quintile—4th quintile	0.70	0.0001
Usual source of care		
Private emergency care—Other	0.41	0.0017
Public emergency care—Public primary care	1.42	0.0106
Public outpatient hospital care—Public primary care	1.64	0.0001
Private emergency care—Public primary care	0.60	0.0502
No USC—Public primary care	1.97	<0.0001
Private emergency care—Public secondary care	0.35	0.0002
Private emergency care—Public emergency care	0.42	<0.0001
No USC—Public emergency care	1.39	0.0245
Private emergency care—Public outpatient hospital care	0.36	<0.0001
Private emergency care—Private outpatient care	0.49	0.0001
No USC—Private outpatient care	1.62	<0.0001
No USC—Private emergency care	3.31	<0.0001

* All *p* values reported have already been adjusted by the Bonferroni method; therefore, if an adjusted *p* value ≅ 0.05, the unadjusted *p* value ≅ 0.0009.

**Table 6 ijerph-19-01829-t006:** Results of the survey-weighted logistic regression model with the main effects of socioeconomic status on dual use of healthcare, controlling for demographic, health needs, and healthcare use variables—PNS 2019 (*n* = 41,921).

Variable	OR	99% CI
Dwelling		
Urban	-	(ref)
Rural	1.24	1.03–1.47
Sex		
Male	-	(ref)
Female	1.06	0.971–1.15
Race		
White	-	(ref)
Non–white	1.04	0.95–1.18
Age		
0–14	-	(ref)
15–24	1.05	0.78–1.42
25–34	1.12	0.83–1.51
35–44	1.06	0.82–1.39
45–54	1.23	0.94–1.60
55–64	1.22	0.92–1.62
65–74	1.18	0.89–1.57
+75	1.49	1.10–2.00
Material wealth		
1st quintile	-	(ref)
2nd quintile	1.15	1.01–1.31
3rd quintile	1.16	0.97–1.37
4th quintile	1.30	1.06–1.61
5th quintile	1.17	0.90–1.53
Income		
1st quintile	-	(ref)
2nd quintile	1.47	1.16–1.86
3rd quintile	2.16	1.71–2.73
4th quintile	2.43	1.91–3.08
5th quintile	1.69	1.26–2.26
Education		
No education/primary incomplete	-	(ref)
Secondary incomplete	1.06	0.88–1.28
Undergraduate incomplete	1.16	0.98–1.39
Undergraduate complete and above	0.98	0.77–1.24
Region		
North	-	(ref)
Northeast	1.02	0.85–1.23
Southeast	0.79	0.64–0.99
South	1.05	0.83–1.32
Center–west	0.78	0.63–0.98
FHS		
Not enrolled/does not know	-	(ref)
Enrolled	1.10	0.95–1.26
Health status		
Very good or good	-	(ref)
Fair	0.88	0.77–1.01
Bad or very bad	0.77	0.62–0.95
Usual source of care		
Other	-	(ref)
Public primary care	0.683	0.45–1.04
Public secondary care	1.15	0.62–2.14
Public emergency care	0.970	0.61–1.54
Public outpatient hospital care	1.12	0.70–1.79
Private outpatient care	0.832	0.54–1.29
Private emergency care	0.406	0.23–0.72
No USC	1.35	0.88–2.07

OR: odds ratio, considering the complex survey design; CI: confidence interval.

## Data Availability

Publicly available datasets were analyzed in this study. These data can be found here: https://www.ibge.gov.br/estatisticas/sociais/saude/9160-pesquisa-nacional-de-saude.html?=&t=microdados (accessed on 5 August 2021).

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
