# Peer review of "Dual Use of Public and Private Health Care Services in Brazil"

_ijerph, 2022, doi:10.3390/ijerph19031829_

Round 1

Reviewer 1 Report

Thanks for the opportunity to review this interesting manuscript! This study aimed to give a description of the dual use of health care in Brazil and to investigate its relationship with socioeconomic status. Dual use was found to happen both on the direction of public to private, and on the direction of private to public. The findings confirm the authors hypothesis. The manuscript is original, scientifically sound and interesting to the readers globally. This is a highly relevant theme in the field of public health management. Below I put some comments to improve this manuscript in order to better reach our global audience.

INTRODUCTION

  • The introduction is well written, and all the key concepts are presented.
  • The authors should be consistent in using the acronym SES (socioeconomic status) in all sections of the manuscript.

MATERIALS AND METHODS

  • Line 105 - authors must exclude the word "anno" or replace it by the word "year"
  • It is clear that this is a secondary study that used Brazilian open data. However, it is unclear whether the authors participated in the Brazilian National Health Survey data collection or not, according to item 2.1. I believe they did not, but in some parts of the text it seems that they did. So maybe it is worth making this statement.

RESULTS

  • Line 262 - “Table 3 presents the ANOVA table or the first model, with main effects only, in which usual source of care, income, and region achieved the 0.09% level of significance according to the Bonferroni-corrected 264 FWER.” ->
  • I believe that Bonferroni correction is not well described in the results as a whole. In table 3 the p-value is described, but it does not show in which groups the differences are. For example in the “Income” group, there are five different groups. Where are the differences, according to the Bonferroni correction? This can be solved by adding this information in the legend of the table or simply describing it in text in the results section.

DISCUSSION

  • The discussion is well written, and the interpretation of the results is appropriate.
  • The authors had a clever solution by proposing in the methodology the definition of dual use in this manner. However, in the PNS survey the source of care was only considered the last care episode. This flaw is understandable because of the design of the National survey but is not something that can be changed by the authors. However, I hypothesize that the results of dual use could be massively higher if the question considered the source of care in the last 12 months, for example. In my opinion you could include this concept in the discussion. This could strengthen your results even more.
  • There are some characteristics of the Brazilian healthcare system that could be better defined for the understanding of a global audience. I put as a suggestion the following points:
  • a) certainly the difference between health insurance systems has an impact on these results, and you have described some of them well. However, I think the differences between the health insurance possibilities (full coverage? partial coverage with financial participation by the user? complementary exams-only coverage?) could be described. These particular characteristics of Brazilian health system can affect the readers' interpretation of the results.
  • b) The authors pointed out in the discussion that “Specialist care is a well-known bottleneck in SUS, being mentioned in some studies as a reason for seeking private care” (Line 382). However, the reverse path is very real, as you pointed out in the introduction: “On the other hand, private users can seek SUS for high-complexity treatments not covered by their plans, medications, and for homecare follow-up with public primary care” (Line 92). In my opinion this concept should be very much strengthened in the discussion: the path of people seeking specialized tertiary care in SUS that is often not available even in the private health care system. I believe that this phenomenon is strongly linked to the results you have found.
  • Line 327 - Is it clear that further research is necessary to clarify the underlying mechanisms of the results. I understand that it was not the aim of this paper to explain the reasons for the results found, but only to describe them. However, as experts on the subject, the authors could stress more of their own opinion on the possible reasons for dual use in this population. This was done in a very discreet way in the discussion and left one wanting to better understand possible underlying mechanisms in the authors' view. You will certainly find support within the references that you have used in this manuscript.

Additional comments:

  • Because this is a secondary study using public data, it is highly reproducible and the flaws are well described by the authors in the "study limitations" section. 
  • Since this is a secondary study in which authors used data from a public database, I believe that Informed consent was obtained from the Brazilian National Health Survey and not from their original study. If this is correct, the Informed Consent Statement (line 453) should be modified.
  •  Self-citations made by authors are relevant to the content of the manuscript
  • Please correct references 33 and 46 by adding the URL

Reviewer 2 Report

This is a very well researched and very well written paper that finds the many Brazilians seek care from a source that is different from their “regular” healthcare source. Brazil has universal public healthcare and an active private health insurance market.  in terms of quality, this paper could be published essentially as is. It uses a very robust source of data, meticulously applies a range of research methods and produces conclusions that are supportable. The only question that I might have is whether this work has universal applicability or whether the situation here is unique to Brazil. I would like to see the authors persuade us about generalizability. For me, the quantitative methodology applied to this study is exemplary.

My only real issue is that overall there is so much here that the paper is a bit overwhelming for a single reviewer. I wonder whether readers might find it overwhelming as well. It does seem that there is enough here for several papers.

In the introduction there is an observation that dual public and private systems are common in low- and middle-income countries. From my experience there are dual systems and a lot of developed countries as well.

The description of the Brazilian healthcare system is pretty lengthy, and I wonder how much of it is really necessary as an introduction to the problem that this paper studies.

The study uses data from the Brazilian National Health Survey. It employs a “survey weighted logistic regression” to look at the Association of SES with dual use of healthcare.

The researchers define dual use as no private health insurance but using private source of healthcare or having private insurance but using a public source. They identify potential variables of interest and used a weighted Principal Component Analysis to reduce the number of variables to one, extracting the first component and splitting into quintiles. It might be worth explaining why the component was split into quintiles.

Only people who sought care in the due weeks prior to the survey were selected for analysis.  The researchers conducted a weighted chi-square analysis to compare the group selected to all survey respondents. They also used a chi-square test to compare the population of dual users to the population of non-dual users.

Survey -weighted binomial logistic regression models were used to evaluate dual use. An ANOVA was performed to determine which variables had an impact on dual use. I am curious why the researchers used ANOVA. The researchers built three models. The third model was developed in six steps.  The third model it seems was developed using “variable mining” of a sort. Social scientists commonly use this approach, but it makes econometricians nervous.

The chi-square test found overrepresentation of white individuals and those of higher socioeconomic status.  Some regions were overrepresented. Private usual sources of health and public emergency care were also overrepresented. The p values for these are all significant at p<.0001. This is to be expected a very large data set like this one.

Dual users were predominantly female, nonwhite, did not have health insurance and were from the southeast region of Brazil. The majority sought care in private facilities (67.3%) but one third of the group seeking care in public facilities is pretty meaningful.

Migrating from one healthcare payer to another increases with material wealth and income quintiles. People 45 and older were overrepresented. Most dual users referred no usual source of care followed by public primary care as their usual source.

The researchers used three multinomial logistic regression models to explore the effect of socioeconomic status, healthcare use and demographic variables and then used a type 3 ANOVA.

Not a lot jumps out of Table 4 in terms of odds ratios with the exception of urban versus rural (OR=1.24), age greater than 74 (OR=1.49), fourth quintile of material wealth (OR=1.30) and the second through the fifth quintiles of income. There’s probably a reason for breaking wealth and income into quintiles but it’s not intuitive to me and I can’t recall ever seeing it. No usual source of care also had an OR of 1.35.  I would have liked to have seen McFadden’s pseudo R square for the models which can be programmed in R.

The discussion notes that women seek care more often than men. This may be a global phenomenon. The age-related variable may reflect greater need for medical care with age, including dual use. The researchers report higher odds of dual use with rural dwellings. The odds ratio is 1.24,. The southeast region and to the center-West region showed a pretty strong “protective” effect for dual use. The researchers spent a substantial portion of the discussion trying to explain the results.

The strong relationship between income and dual use prompted the researchers to raise issues about health inequity and over use of health care. Another finding was that main facilities sought by dual users or private outpatient care and public primary care.  The researchers also discuss their interaction terms and the likely rationale for their significance.

The researchers appropriately note that chi-square tests are sensitive to sample sizes which may lead to new differences being detected as significant. I think that they are correct and that may be a significant limitation here. They did conduct a Bonferroni correction to attempt to deal with this.

The conclusion reached by the authors that the research has important insights for policymakers is supported by their findings and correct.

Reviewer 3 Report

I recommend that the authors insert illustrative figures of the results obtained, the tables are in some cases difficult to understand.

Author Response

Reviewer 3

I recommend that the authors insert illustrative figures of the results obtained, the tables are in some cases difficult to understand.

  • One figure with 3 panels has been added on line 266 to show the distribution of dual use across different SES levels.

Reviewer 4 Report

Dear Dr. Dear Dr.Silva and Dr. Willems,

Thank you for your contribution to this manuscript,

However, there are still some places that can be improved.

Since it is a cross-section study, let's go over the AHRQ form first (in the attachment).  

  1. Could you give specific time period used data from the Brazilian National Health Survey anno 2019(line 105)?

Here is the example.

“The present study involved a secondary analysis of MDHS data (collected between December 2015 and July 2016).”

Nozaki, Ikuma, Masahiko Hachiya, and Tomomi Kitamura. "Factors influencing basic vaccination coverage in Myanmar: secondary analysis of 2015 Myanmar demographic and health survey data." BMC public health 19.1 (2019): 1-8.

  1. Please Indicate if evaluators of subjective components of the study were masked to other aspects of the status of the participants(double-blind procedure?) If you did not do, please explain or write this part the limitation paragraph.
  2. Please add a flow chat for participants of the study, including the number of patients in each cluster and the cases with missing data/ response rates and completeness of data collection.

Here is the example (Figure 1 in the article below).

Ye, Wenjing, et al. "Identification of COVID-19 clinical phenotypes by principal component analysis-based cluster analysis." Frontiers in medicine 7 (2020): 782. 

  1. Could you list the definitions of the terms in a table? (to summarize the definition of dual use and variable coding part)
  2. The highlight of this article is the “Survey-Weighted Logistic Regression Model”. That is why you also keep the forward and backward model in supplementary data(Line 395). How about comparing the IC99% from these two models in a Forest Plot?

Figure2 in the article below is an example.

Nikiforuk, Aidan M., et al. "Influence of chronic hepatitis C infection on the monocyte-to-platelet ratio: data analysis from the National Health and Nutrition Examination Survey (2009–2016)." BMC public health 21.1 (2021): 1-11.
